# Spatial Effect Analysis of Forestry Technology Innovation on Forestry Industry Economic Growth

**Zhuoya Ma** [1], **Tianle Liu** [1], **Jing Li** [1], **Zhentao Liu** [2] **and Wenhui Chen** [1,*]

1　School of Economics and Management, Beijing Forestry University, Beijing 100083, China
2　School of Economics and Management, Hebei Agricultural University, Baoding 071001, China
*　Correspondence: wenhui@bjfu.edu.cn

**Abstract:** The forestry industry is a low-carbon green industry with great potential for development, but its current development model is facing multiple dilemmas that urgently require a shift to high-quality forestry development. Neoclassical theory and endogenous growth theory consider technology innovation as the foundation of economic growth. In order to explore the impact of forestry technology innovation (FTI) on the economic growth of forestry industry, this paper uses the entropy method, spatial Durbin model, and threshold model to explore the spatial effect of forestry technology innovation on forestry industry economy for analysis and exploration of the non-linear relationship between the two, and the panel data from 31 provinces in China from 2002 to 2020 are used as the sample for empirical study. Results show that: (1) Forestry technology innovation has a significant role in promoting the economic growth of the forestry industry and has obvious spatial spillover effects, which also promotes the economic growth of forestry in neighboring regions. (2) There is a threshold effect of forestry technology innovation on the economic growth of the forestry industry. When the forestry technology innovation exceeds its optimal interval, its effect on the economic growth of forestry industry shows diminishing marginal benefits. (3) Forestry technology innovation has industrial heterogeneity on the economic growth of the forestry industry. Therefore, managers should take advantage of the growth effect of forestry technology innovation in the forestry industry at the level of sustainability, and further make full use of the spatial effect resulting from the flow of technology to establish a system of communication and learning to form a virtuous coordination of the forestry economic environment for the high-quality development of the forestry industry.

**Keywords:** forestry technology innovation; forestry industry economic growth; spatial Dubin model; threshold model; spatial effects

## 1. Introduction

　　The forestry industry is a material production and ecological construction project that protects, cultivates, manages, and utilizes forest resources and provides forest products as well as forest services to society [1]. The forestry industry is a green and low-carbon industry with great potential for development, and it is an important area of economic development in countries around the world [2]. The development of the forestry industry not only promotes the construction of forest areas and the profit of community farmers [3], but also relates to the sustainable economic and social development of countries around the world. However, with the rising global temperature, degradation of natural environment, natural resource constraints, and declining environmental carrying capacity [4], how the forestry industry should change the original development model, improve the efficiency of the forestry industry, and achieve high-quality forestry development has become the focus of scholars worldwide. China is one of the major forestry industry countries [5,6]. After decades of development, especially the establishment of the socialist market economy system, the development of China's forestry industry has advanced rapidly and attracted

the attention of many scholars. At present, China's forestry construction is entering a critical period, and the problems of natural resource constraints, elevated human capital, and low returns on investment are exceptionally severe, so solving these problems can provide important lessons for the healthy development of the forestry industry [7]. Therefore, this paper will explore the impact of forestry technology innovation on the economic growth of the forestry industry based on China's forestry industry panel data, in order to provide a reference for the development of forestry industry all over the world. According to official data from the National Bureau of Statistics of China, it can be seen that the total output value of China's forestry industry grew from 10.39 billion USD in 1949 to 11,768.73 billion USD in 2020 (These data eliminate the effect of inflation and are converted based on the 1949 RMB to USD exchange rate), the total output value of forestry increased 1096 times [8,9]; from its proportion of GDP, it grew from 3.71% in 1952 to 8% in 2020. See Appendix A (Table A1) for specific forestry industry classifications.

For a long time, China's forestry development mainly relied on increasing input factors to bring economic growth, and this forestry economic development model needs to be improved urgently. Thus, the National Forestry and Grassland Administration of China has formulated a forestry development plan for the next five years in 2017 and 2021, respectively, issuing a series of policies to improve the efficiency of the forestry sector and promote forestry modernization [10]. Currently, forestry construction is entering an important period of transformation, while technology innovation is the key [11]. Technology innovation is not only a critical initiative to improve the efficiency of resource utilization and transform the crude development mode, but also an important tool to explore the quality of economic growth and the contribution of technological progress to economic growth. With the increasing resource and environmental constraints in recent years, the contribution of technology innovation to forestry economic growth has become increasingly obvious, especially in improving labor productivity and product quality [12]. Therefore, when exploring the impact of forestry technology innovation on the forestry industry economy, it is crucial to accurately measure the level of forestry technology innovation. It can also exploit new technologies and methods to achieve improvements in seeds, fertilizers, mechanical equipment and reduce the risk to economic growth, which in turn can improve productivity development [13,14]. However, China's forestry technology innovation ability is relatively weak compared with developed countries. Meanwhile, due to the limitation of funds and institutional constraints, the role of forestry technology innovation in boosting forestry economic growth has not been fully explored, and it is difficult to play the advantages of technology innovation [15]. Although the role of technology innovation in boosting forestry economic growth has become a common sense in reality [16], there is a shortage of empirical studies to accurately estimate the role of forestry technology innovation on the economic development of forestry industry. To answer the above questions, it is necessary to sort out the relevant theories and combine them with the realistic background for empirical studies. Therefore, research on the role of forestry technology innovation on forestry economic growth has gained the attention of scholars. Forestry technology innovation is important for forestry industry economy, but human capital, industrial structure and investment are also important factors affecting forestry industry economy, and scholars have done in-depth analysis on this [17,18]. Moreover, the impact of forestry technology innovation on the economy of forestry industry has not been clarified. This paper tries to control other influencing factors in the case of the best effort to clarify the level of forestry technology innovation in China at this stage. Therefore, this paper, while trying to control other influencing factors, clarifies the level of forestry technology innovation in China at this stage, explores the driving effect of forestry technology innovation on forestry economic growth, and explores the relationship between them, so as to better realize the rapid growth and transformation of the forestry industry and provide reference values for achieving the high-quality development of global forestry.

Reviewing the literature, domestic and foreign scholars are more concerned with forestry industry agglomeration and structure optimization and adjustment, forestry indus-

try performance evaluation, forestry production efficiency, and high-quality development of forestry industry [19–22], as well as elaborating the influencing factors and methods of forestry economic development [23,24]. At the same time, some scholars [25,26] introduced technology innovation into agricultural production and empirically tested that technology innovation can promote a sustained increase in China's agricultural economy. While Loft et al. [27] explored the positive role played by innovation in the governance of forest ecosystem services from a qualitative perspective, other scholars [28] have explored the impact on human well-being and global sustainable development from an ecological perspective based on technological innovation. Razminienė K et al. [29] studied the competitiveness of clusters through knowledge transfer and innovation, which in turn contribute to regional development. Kovacova M et al. [30] studied intelligent process planning for sustainable organizations in production, capable of tracing industrial products; while Durana P et al. [31] analyzed and estimated the decision process in sustainable smart manufacturing. Zvarikova K et al. [32] further explored the use of artificial intelligence based on decision algorithms for assessing the performance of the sustainability industry, laying the groundwork for future assessments of the development performance of the forestry industry. This all shows that the progress of modern technology to promote the process of industrial production, reflecting the development of the forestry industry can't be separated from the promotion of technology and innovation.

The research examining the economic impact on forestry industry from the perspective of forestry technology innovation is very limited [33,34]. Most of the existing impact analysis of regional forestry development is based on the premise that each region is independent of each other, and spatial spillover effects are not taken into account. The spatial spillover effect is one of the important features of forestry technology innovation activities. The spillover effect occurs when a region benefits from the technology innovation of other regions without having to bear the cost of innovation. This effect is expressed by the fact that technology innovation not only has an impact on local production, but also has a radiating effect on the production activities in the surrounding areas through spatial diffusion [35]. Moreover, the impact of forestry technology innovation on forestry industry production is not linear; the two have different thresholds in the dynamic changes, which creates a threshold effect [36]. When a large investment in forestry technology innovation exceeds the threshold value, it will lead to human capital enrichment, which will produce the "crowding out effect", i.e., the spillover of talents and resources [37]. It can be seen that there are insufficient studies on the impact of forestry technology innovation on the economic growth of the forestry industry and the spatial economic effect. Besides, the research on how this spatial effect affects the economy of the forestry industry and the impact mechanism still needs to be improved.

This paper mainly studies the following questions: (1) At the same time, what is the impact of forestry technology innovation on forestry economic growth? Does it promote the development of the forestry industry economy? (2) Is there a spatial spillover effect? (3) Is there any non-linear relationship between the two? (4) How can we effectively use the positive impact of forestry technology innovation to achieve the sustainable growth of the forestry economy?

The academic contributions of this paper include: (1) In terms of research content, the entropy method is used to measure technology innovation and use the spatial Durbin model to explore the changes in its impact and marginal contribution to the forest industry economy, as well as to analyze the impact factors, in order to provide ideas for upgrading the forestry industry structure and forestry economic transformation. (2) In terms of research methodology, although the spillover of forestry technology innovation exists in adjacent areas with a clear geographical component, it is ignored in actual studies, which may lead to inaccurate findings and underestimate the contribution of forestry technology innovation levels. Therefore, this paper makes up for the shortcomings of the existing studies, in which each region exists independently of the other. This paper extends the impact of forestry technology innovation on the forestry industry economy

to the spatial dimension, fully considering the spatial spillover effect to better explore the spatial correlation effect of forestry technology innovation.

## 2. Research Hypothesis

### 2.1. The Impact of Forestry Technology Innovation on the Economy of Forestry Industry

Forestry technology innovation is the first productive force to promote the economic development of forestry industry, which can not only improve the production efficiency, but also influences the economic growth of the forestry industry. Therefore, it is crucial to accurately identify the role of forestry technology innovation on the forestry economy to enhance the level of forestry technology innovation and further promote the development of forestry economy. Through the previous research, it can be seen that the achievements of forestry technology and innovation can improve the types of forest products, enhance the quality of forest products, improve labor efficiency, and form scale effects [38]. They can also integrate forestry resources as well as reduce costs and resource losses, thus maximizing the benefits of forestry economy and ecological environment. Therefore, Hypothesis 1 is proposed in this paper.

**Hypothesis 1.** *Forestry technology innovation can significantly contribute to the economic growth of the forestry industry.*

### 2.2. The Local-Neighborhood Effect of Forestry Technology Innovation on Forestry Industry Economy

Forestry technology innovation has shared and public good qualities [39]. It can promote the economic growth of forestry industry in the region, improve the utilization rate of resources and the quality of forest products, reduce the damage to the ecological environment in the process of industrial development, and realize the coordinated development of the forestry industry economy and ecological environmental protection. Moreover, forestry production is an open system with social interactions between farmers, especially between different production agents in close proximity to areas with similar natural conditions that would transfer and imitate the application of scientific knowledge [40]. Farmers use technology diffusion for communication, learning, and interaction, which ultimately affects their decision-making behavior. This is consistent with the findings of Anselin [41] et al. that there is a spatial spillover effect of technology innovation in driving economic development. Therefore, Hypothesis 2 is proposed in this paper.

**Hypothesis 2.** *Forestry technology innovation has a positive impact on the economic growth of the local forestry industry, and there is a spatial spillover effect on neighboring areas.*

### 2.3. The Threshold Effect of Forestry Technology Innovation on the Economic Growth of Forestry Industry

The rapid development of the forestry economy is conducive to the formation of forestry industry agglomeration, which can provide a better working platform and more development opportunities, and accordingly attract more outstanding talents. Then, this is reversed to promote the improvement of forestry technology innovation and promote the economic development of forestry industry, ultimately forming a virtuous cycle [42]. However, in the case of limited resources, the forestry technology innovation and talent enrichment to a certain level will result in undesirable competition and a waste of resources [43]. That is, the high concentration of resources not only causes waste, but also makes it difficult to fully exploit the role of forestry technology innovation in promoting the economic growth of forestry industry [44,45]. Therefore, a reasonable agglomeration effect of forestry technology innovation can better play the role of promoting forestry economic growth. Therefore, hypothesis 3 is proposed in this paper.

**Hypothesis 3.** *The impact of forestry technology innovation on the economic growth of the forestry industry has a threshold effect.*

## 3. Data Sources and Methodology

### 3.1. Measurement of Forestry Technology Innovation Level

3.1.1. Parameters

Technology innovation refers to some new technologies and methods invented and applied to the real forestry industry production and practice based on the environment and socio-economic development, thus achieving the goal of improving economic and ecological benefits. Through previous studies, some domestic scholars [46,47] have found that using only the number of patents in measuring the level of technology innovation has greater limitations, pointing out that this indicator can only indicate a certain aspect of technology innovation and cannot fully measure the complex issue of the level of technology innovation, and other more input and outcome variables should be fully considered to improve the scientific and accurate measurement of technology innovation, etc. Therefore, combining the research contents of previous scholars [43,48–50], this paper constructs an evaluation index system of forestry technology innovation based on three dimensions of innovation base, innovation input, and innovation output, which mainly includes three aspects. (1) The foundation of forestry technology innovation, i.e., the economic and social development status, human capital, and other material infrastructure and related facilities of the region where the main body of forestry technology innovation is located, providing key support for technological innovation. (2) Technology innovation input, i.e., the full application of various innovative knowledge and resources in forestry technology activities and technology transformation, is an important part of the integration and distribution of the use of technology innovation. (3) Technology innovation output, i.e., the part of the output and benefit increase of forestry technology innovation achievements, which can test the results and productivity levels of forestry technology innovation, and this is the final result of forestry technology innovation (Table 1).

**Table 1.** Evaluation index system of forestry technology innovation level.

| Indicator Layer | Indicator Type | Measurement |
|---|---|---|
| Innovation foundation | Economic | GDP (billion yuan) |
| | Human capital | Number of higher education institutions (units) |
| Innovation inputs | Labor | The number of enterprises, institutions and agencies by industry in each region (units) |
| | | Number of forestry system employees at the end of the year (person) |
| | | Research and experimental staff (person-years) |
| | Investment | Forestry and grass investment completed this year (million yuan) |
| | | Funding for research and experimental activities (million yuan) |
| | | Forestry technology activities funding (million yuan) |
| Innovation output | Economic output | Per capita income of farmers (yuan) |
| | Patent output | Number of domestic patents granted (pcs) |

3.1.2. Measurement Results

This paper uses the entropy value method to measure the level of forestry technology innovation in 31 provinces in China from 2002 to 2020, respectively, and calculates the average value of forestry technology innovation level in 31 provinces from 2002 to 2020. Then, the line graphs were drawn by selecting the forestry technology innovation data of the study sample in 2002, 2011, and 2020 and the mean data of the study sample. Based on the trends of the line graphs, a preliminary analysis of the basic direction and magnitude of changes of forestry technology innovation level in 31 provinces of China from 2002 to 2020 and the unevenness of forestry technology innovation level in different regions of China was

conducted. Figure 1 shows the forestry technology innovation level of 31 provinces in 2002, 2011, and 2020 and the average value of forestry technology innovation of 31 provinces in China from 2002 to 2020. On the whole, the level of forestry technology innovation in China is uneven, and the level of forestry technology innovation in the eastern and northeastern regions shows a decreasing trend overall, while the level of forestry technology innovation in the central and western provinces shows a fluctuating upward trend. In 2002–2020, in terms of time, the level of forestry technology innovation in 2020 increased by 10.88% compared to the same period last year. The annual growth rate of 33.7% in 2008 was the highest in history, and then declined each year before steadily increasing. The forestry technology innovation level above the average accounted for 68.4% of the total number of years. From a regional perspective, the eastern region has the highest level of forestry technology innovation, followed by the central and northeastern regions, while the western region has the slowest [51]. Moreover, the forest area varies greatly among regions in China, as 14 regions have more than 50% of their land area covered by forests, 14 regions have between 10% and 50% of their land area covered by forests, and two regions have less than 10% of their land areas covered by forests. Specifically, the areas with high levels of forestry technology innovation are mainly concentrated in industrial agglomerations with fast economic development and relatively abundant forestry resources, such as Guangdong, Heilongjiang, and Sichuan, which create good economic and ecological conditions for forestry technology innovation [52,53]. For example, the Guangdong region, in addition to having huge forest resources, also has the advantage of complete industry, which lays the foundation for the development of the forestry industry.

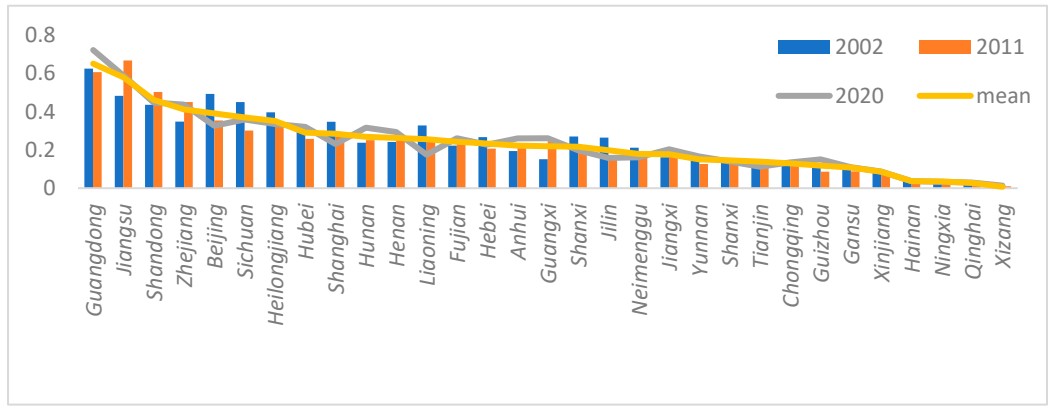

**Figure 1.** Forestry technology innovation levels of 31 provinces in 2002, 2011 and 2020.

*3.2. Moran's I Index*

The Moran index is usually used to test whether there is spatial dependence and spatial autocorrelation, so this paper first verifies the indicator data before using the spatial measurement method. The specific calculation formula is as follows:

$$Moran's\ I = \frac{\sum_{i=1}^{n}\sum_{j=1}^{n}w_{ij}(x_i-\overline{x})(x_j-\overline{x})}{s^2\sum_{i=1}^{n}\sum_{j=1}^{n}w_{ij}} \tag{1}$$

where $x_i$ is the $i_{th}$ attribute value, $\widetilde{x}$ is the mean of indicator $x$, and $s^2$ is the variance of indicator $x$, Where $\widetilde{x} = \frac{1}{n}\sum_{i=1}^{n}x_i$ , $s^2 = \frac{1}{n}\sum_{i=1}^{n}(x_i-\widetilde{x})^2$. $w_{ij}$ is the spatial weight matrix, $w_{ij}$ is the $i_{th}$ row and $j_{th}$ column of the spatial weight matrix $w$; $n$ represents the number of regions, $i = 1, 2, \ldots , T, j = 1,2, \ldots , N$. The following spatial weight matrix is selected in this paper: firstly, the assignment is based on whether there is a spatial neighborhood relationship between region $i$ and region $j$. If neighboring, it is 1, and if not, it is 0. In turn, the neighboring spatial weight matrix is obtained. Then, the inverse of the road distance between two areas is used to construct the geographic distance weight matrix [54], when

the road distance between two areas is shorter, the weight is larger. Finally, the absolute value of the reciprocal per capita GDP difference of each region is selected to construct the economic weight matrix, and the more similar the economic level of the two regions, the greater the weight. With *Moran's I* values between −1 and 1, the larger the absolute value, the higher the spatial autocorrelation, the closer to 0 indicates that the regional attributes are randomly distributed and there is no spatial correlation. When *Moran's I* > 0, this indicates positive correlation; when *Moran's I* < 0, this indicates negative correlation [6].

*3.3. Spatial Durbin Econometric Model*

3.3.1. Spatial Model Construction

To study the impact of forestry technology innovation on the economic growth of forestry industry under the spatial spillover effect, spatial variables will be introduced into the model and a general spatial nested model will be constructed first.

$$EGFI_{it} = \delta W_{ij} EGFI_{it} + FTI_{it}\beta + W_{ij} FTI_{it}\theta_1 + X_{it}\theta_2 + u_{it}, \ u_{it} = \lambda W_{ij}u + \varepsilon_{it} \quad (2)$$

where EGFI is the economic growth of forestry industry and FTI is the forestry technology innovation, $W_{ij}$ is the $i_{th}$ row and $j_{th}$ column of the spatial weight matrix $W$, $\delta$ is the coefficient of the spatial lag term of the explained variable. $\beta$ is the explanatory variable regression coefficient, $\theta$ is the spatial autocorrelation coefficient of the explanatory variable, $\varepsilon$ represents the disturbance term, and $\lambda$ is the coefficient of the error term, $i = 1, 2, \ldots ,$ T, $j = 1,2, \ldots ,$ N. $W$ represents the spatial weight matrices, and $W_{ij}$ represents the spatial weight matrices of city $i$ and city $j$.

When the following conditions are satisfied, Formula (2) is a spatial lag model (SAR).

$$EGFI_{it} = \delta W_{ij} EGFI_{it} + FTI_{it}\beta + \varepsilon_{it}; \ If \ \lambda = 0, \ \theta = 0 \quad (3)$$

When the following conditions are satisfied, Formula (2) is a spatial error model (SEM).

$$EGFI_{it} = FTI_{it}\beta + u_{it}, \ u_{it} = \lambda W_{ij}u + \varepsilon_{it} \ ; \ If \ \delta = 0, \theta = 0 \quad (4)$$

When the following conditions are satisfied, Formula (2) is a spatial Durbin model (SDM).

$$EGFI_{it} = \delta W_{ij} EGFI_{it} + FTI_{it}\beta + W_{ij} FTI_{it}\theta_1 + X_{it}\theta_2 + \varepsilon_{it}; \ If \ \lambda = 0 \quad (5)$$

To prevent bias and endogeneity problems caused by model estimation, this paper constructs SAR, SDM, and SEM spatial panel data models, as shown in Equations (3)–(5), respectively, and then selects the appropriate models based on LM test, LR test, Wald test, and Hausman test [48].

3.3.2. Threshold Panel Model

To study the relationship between forestry technology innovation and the economic growth of the forestry industry, the following threshold model is built. Equation (6) is a single threshold model and Equation (7) is a double threshold model.

$$EGFI_{it} = u_i + \beta_1 FTI_{it} * I(FTI_{it} \leq \gamma) + \beta_2 FTI_{it} * I(FTI_{it} > \gamma) + \varepsilon_{it} \quad (6)$$

$$EGFI_{it} = u_i + \beta_1 FTI_{it} * I(FTI_{it} \leq \gamma) + \beta_2 FTI_{it} * I(\gamma_1 < FTI_{it} \leq \gamma_2) + \beta_3 FTI_{it} * I(FTI_{it} > \gamma_2) + \varepsilon_{it} \quad (7)$$

where EGFI, $\varepsilon$ and FTI are as above, $q_{it}$ is the threshold variable, $\gamma$ is the threshold value, and $I(.)$ is an indicator function of the threshold variable.

*3.4. Data Sources*

This paper selects panel data of 31 provinces from 2002 to 2020 for the study, and the data are obtained from China Statistical Yearbook, China Forestry Statistical Yearbook, China Rural Statistical Yearbook, China Science and Technology Statistical Yearbook, rele-

vant statistical yearbooks of each province, and EPS database. There are two main methods to deal with missing data in this paper, one is to use the moving mean method, which is to take the average of two adjacent years, and the other is to use linear interpolation to fill in.

*3.5. Variables*

Explained variable. The level of economic growth of forestry industry (EGFI): the increase of output is the most intuitive expression of economic growth of forestry industry. This paper takes the forestry output value as the explained variable, which contains the results and benefits of the forestry industry economy in a certain period of time and is a more systematic and comprehensive output indicator of forestry industry [55].

Core explanatory variables. The level of forestry technology innovation (FTI): In previous literature, scholars have used the number of patent applications, authorizations, and sales revenue after new product development to measure. Due to the lack of relevant data on FTI, this paper constructs an evaluation index system from three dimensions of forestry innovation input, innovation output, and innovation foundation, and uses the entropy value method to measure them [56].

Control variables. Financial expenditures of the government (FEG): government financial support can have an impact on the economic development of the local forestry industry. In this paper, the share of financial expenditures of foresters in the gross local product of each province was selected to measure the degree of economic participation in the forestry industry in that province [26]. Forestry fixed asset level (FFL), i.e., the level of investment in forestry fixed assets, can directly affect the economic level of forestry industry and create favorable conditions for the economic growth of the forestry industry, which is expressed in this paper using the amount of investment in forestry fixed assets [25]. Disaster prevention rate (DPR), i.e., the occurrence of disasters, will directly affect the output value of forestry industry, among which pests and diseases will directly affect the economic growth of forestry. In this paper, the control rate of forest pests and diseases is selected to characterize the disaster control rate. Education level (EL): the higher the level of human capital's education, the faster it accepts new knowledge, which in turn enhances FTI capacity and promotes the process of forestry economic development [57,58], which is represented in this paper by the number of general undergraduate students in higher education institutions. Upgrading of industrial structure (UIS): in this paper, we choose to use the proportion of the total output value of the secondary and tertiary industries to the total regional output value to characterize [59]. Land area (LA): the amount of forest area will directly affect the output value of forestry, so this paper selects forest area to characterize land input. Descriptive statistics for each variable are listed in Table 2.

**Table 2.** Descriptive statistics of variables.

| Type | Variable | Mean | Std. Dev. | Min | Max |
|---|---|---|---|---|---|
| Explained variable | EGFI | 0.1045 | 0.1008 | 0.0012 | 0.4374 |
| Core explanatory variables | FTI | 0.2183 | 0.1500 | 0.0002 | 0.7220 |
| Control variables | FEG | 2.8314 | 3.1931 | 0.3475 | 26.2107 |
| | ln FFL | 11.3793 | 1.6191 | 4.1897 | 16.04661 |
| | DPR | 72.8102 | 22.1586 | 6.0000 | 100.002 |
| | EL | 13.1112 | 1.031 | 9.0405 | 14.7287 |
| | UIS | 88.1905 | 6.4460 | 59.2343 | 99.7324 |
| | ln LA | 6.3197 | 1.3959 | 0.8109 | 8.4116 |

## 4. Results

*4.1. Spatial Autocorrelation Test*

An autocorrelation test is carried out to determine whether there is spatial correlation among the FTI and forestry industry economic development level occurring in the various regions examined. Table 3 shows the Moran index of FTI and economic growth of forestry

industry in 31 provinces in China from 2002 to 2020 under the neighboring weight matrix, geographic distance weight matrix, and economic weight matrix. It can be seen that the economic growth of the forestry industry and the level of FTI under the three spatial weight matrices basically have a significant positive spatial correlation, i.e., provinces with similar economic growth rate of forestry industry are highly clustered in space [60]. At the same time, forestry output value and FTI level as a whole show a gradual trend of becoming larger over time, indicating that the degree of agglomeration is increasing.

**Table 3.** Global Moran Index based on FTI and forestry output.

| Year | Forestry Technology Innovation | | | Forestry Output | | |
|------|------|------|------|------|------|------|
|      | (1) | (2) | (3) | (4) | (5) | (6) |
| 2002 | −0.032 | −0.055 | 0.071 | 0.057 *** | 0.394 *** | 0.109 * |
| 2003 | −0.050 | −0.092 | 0.089 * | 0.042 ** | 0.288 *** | 0.057 |
| 2004 | −0.054 | −0.100 | 0.064 | 0.050 ** | 0.293 *** | 0.057 |
| 2005 | −0.047 | −0.095 | 0.053 | 0.060 *** | 0.289 *** | 0.048 |
| 2006 | −0.037 | −0.068 | 0.055 | 0.059 ** | 0.259 *** | 0.017 |
| 2007 | −0.051 | −0.105 | 0.042 | 0.052 ** | 0.238 *** | 0.014 |
| 2008 | −0.003 | 0.061 | 0.112 * | 0.055 ** | 0.262 *** | 0.007 |
| 2009 | 0.010 | 0.109 | 0.102 * | 0.054 ** | 0.253 *** | 0.018 |
| 2010 | 0.007 | 0.100 | 0.093 * | 0.084 *** | 0.453 *** | 0.082 |
| 2011 | 0.022 * | 0.153 * | 0.120 ** | 0.084 *** | 0.445 *** | 0.075 |
| 2012 | 0.022 * | 0.154 * | 0.121 ** | 0.079 *** | 0.425 *** | 0.066 |
| 2013 | 0.026 * | 0.161 ** | 0.123 ** | 0.075 *** | 0.378 *** | 0.047 |
| 2014 | 0.027 * | 0.167 ** | 0.112 * | 0.081 *** | 0.387 *** | 0.054 |
| 2015 | 0.024 * | 0.152 * | 0.122 ** | 0.099 *** | 0.419 *** | 0.053 |
| 2016 | 0.021 * | 0.143 * | 0.119 ** | 0.113 *** | 0.458 *** | 0.051 |
| 2017 | 0.015 | 0.122 * | 0.109 * | 0.099 *** | 0.389 *** | 0.081 |
| 2018 | 0.012 | 0.104 | 0.085 * | 0.104 *** | 0.408 *** | 0.069 |
| 2019 | 0.005 | 0.140 * | 0.090 * | 0.107 *** | 0.411 *** | 0.047 |
| 2020 | 0.002 | 0.064 | 0.049 | 0.109 *** | 0.417 *** | 0.050 |

Notes: *** $p < 0.01$, ** $p < 0.05$, * $p < 0.1$; (1)–(3), (4)–(6) represent the three spatial weight matrices under geographical distance, neighboring distance, and economic distance, respectively.

*4.2. Spatial Regression Analysis*

4.2.1. Selection of the Spatial Econometric Model

To determine the type of spatial interaction effect model that should be used, a statistical test on the regression results regarding FTI and forestry industry economy is conducted, and the results are shown in Table 4. First, the Lagrange multiplier (LM) test of the model with significant *p*-value results rejects the original hypothesis that there are spatial errors and spatial lag effects. Second, the Hausman test is applied to determine whether the random-effects model or the fixed-effects model is selected. If the *p*-value of Hausman test statistic is 0, the original hypothesis is rejected and the fixed-effects model is chosen; conversely, the random-effects model is chosen. Furthermore, the original hypothesis was rejected by testing the *p*-value significantly, so the fixed-effects model was chosen, time fixed in both directions was chosen as the baseline analysis model, and the parameter estimation was performed using the maximum likelihood method [52]. Finally, the combination of the likelihood ratio (LR) test and Wald test results basically rejects the original hypothesis at the 1% significant level, so the spatial Durbin model (SDM) cannot be simplified to be used as a spatial lag model (SAR) or spatial error model (SEM), indicating that the spatial Durbin model setting in this paper is reasonable.

**Table 4.** Statistical tests for spatial Durbin model selection.

| Test Parameters | Statistics Results | *p* | Results |
|---|---|---|---|
| LR_lag | 123.28 | 0.0000 | Reject |
| LR_error | 155.59 | 0.0000 | Reject |
| Wald_lag | 50.30 | 0.0000 | Reject |
| Wald_error | 51.83 | 0.0000 | Reject |

4.2.2. Spatial Econometric Benchmark Regression Results

Based on relevant data from 31 provinces across China from 2002 to 2020, this paper included FTI as a core variable in the spatial Durbin model for regression under the spatial weight matrix of geographical distance, and the results are reported in Table 5. According to the results, it can be seen that the regression coefficients of FTI indicators are significantly positive at the 1% level. The higher the level of forestry science and technology innovation, the faster the economic growth of the forestry industry, and its economic meaning is equally significant. Thus, hypothesis 1 was verified. Moreover, the spatial lag term (Wscore) and the spatial autoregressive coefficient Spatial rho are also significant at the 1% level in the table, which confirmed the existence of spatial spillover effect of FTI, and if the spillover effect was ignored it would cause bias in the assessment of the economic benefits of FTI, which also initially verified hypothesis 2.

**Table 5.** Spatial Durbin model parameter estimation results.

| Variable | Main | Variables | Main |
|---|---|---|---|
| FTI | 0.128 *** | W FTI | 1.002 *** |
| | (0.0251) | | (0.121) |
| FEG | −0.0118 *** | W FEG | 0.0172 ** |
| | (0.0013) | | (0.0070) |
| ln FFL | 0.0052 *** | W ln FFL | −0.0040 |
| | (0.0018) | | (0.0114) |
| DPR | −0.0001 | W DPR | −0.0036 *** |
| | (0.0001) | | (0.0011) |
| ln EL | 0.0089 * | W ln EL | 0.0391 * |
| | (0.0048) | | (0.0213) |
| UIS | −0.0021 *** | W UIS | −0.0056 ** |
| | (0.0007) | | (0.0026) |
| ln LA | 0.0272 *** | W ln LA | −0.0035 |
| | (0.0029) | | (0.0163) |
| Spatial rho | 0.5030 *** | Variance sigma2_e | 0.0030 *** |
| | (0.103) | | (0.0002) |

Note: ***, **, and * represent significance levels of 1%, 5%, and 10%, respectively.

From the main effects of the influencing factors of the control variables, forestry fixed asset investment, the number of students in higher education, and forest area are significantly and positively correlated with forestry economic growth, which is consistent with the findings of Li Haipeng et al. [61], while government financial expenditure and industrial structure are significantly and negatively correlated with forestry industry economic growth. Here, the spatial autoregressive coefficient of forestry industry economic growth is 0.503 and significantly positive, and the spatial interaction term coefficient (W score) of FTI also passes the 1% significance level test, indicating that there is an obvious endogenous interaction effect of forestry industry economic growth between regions. In terms of the spatial lag term, the spatial interaction coefficients of the number of students in higher education and government financial expenditure are significantly positive, showing that indicators will have a significant positive impact on the economic growth of the forestry industry through the neighboring geospatial mechanism. In contrast, natural disaster prevention and control rate and industrial structure upgrading have a significant negative

impact on the forestry industry structure in neighboring regions, which is consistent with the findings of Sikora et al. [62].

### 4.2.3. Decomposition of Spatial Effect

According to Le Sage and Pace (2009) and other scholars [63], the existence of spillover effects within a region cannot be judged simply by whether the parameter values of the spatial explanatory variables are significant or not, and the regression results need to be decomposed again by partial differencing. The results of regression are further decomposed in this paper to derive the direct and indirect effects of FTI (Table 6).

**Table 6.** Marginal effect decomposition of spatial Durbin model.

| Variable | Direct Effect | Indirect Effect | Total Effect |
|---|---|---|---|
| FTI | 0.1770 *** | 2.072 *** | 2.250 *** |
| | (0.0294) | (0.469) | (0.485) |
| FEG | −0.0113 *** | 0.0221 * | 0.0108 |
| | (0.0013) | (0.0134) | (0.0139) |
| ln FFL | 0.0054 *** | −0.0008 | 0.0046 |
| | (0.0020) | (0.0227) | (0.0239) |
| DPR | −0.0003 | −0.0072 *** | −0.0075 *** |
| | (0.0002) | (0.0025) | (0.0026) |
| ln EL | 0.0107 ** | 0.0870 ** | 0.0977 ** |
| | (0.0049) | (0.0440) | (0.0458) |
| UIS | −0.0023 *** | −0.0132 ** | −0.0155 *** |
| | (0.0007) | (0.0053) | (0.0056) |
| ln LA | 0.0277 *** | 0.0185 | 0.0462 |
| | (0.0031) | (0.0327) | (0.0346) |

Note: ***, **, and * represent significance levels of 1%, 5%, and 10%, respectively.

According to the decomposition results, it shows that there are spatial economic benefits of FTI, which confirms hypothesis 2. The direct, indirect, and total effects of FTI level under the spatial weight matrix of geographical distance are significantly positive at the 1% level, further verifying hypothesis 2. This result indicates that FTI can not only significantly promote the forestry economic growth in the region, but also significantly promote the forestry economic growth in the neighborhood. When other conditions remain constant, the total output value of local forestry industry will increase by 0.177 for every 1 unit increase in the level of FTI, which also proves that FTI is an important source of power to promote the economic development of forestry industry. Furthermore, FTI can have a positive impact on the development of the local forestry industry economy in different ways, drive the technological revolution and rapid growth of the region, and promote the rapid development of the regional forestry industry economy.

From the decomposition coefficients of the spatial effects of the control variables, (1) the direct effect of the share of financial support to agriculture is significantly negative and the indirect effect is significantly positive, indicating that the government's economic interventions hinder the growth of the forestry economy in the local and surrounding areas, which may be due to the fact that the government's economic interventions make the local forestry development dependent on it, thus not bringing out the expected benefits of financial support to agriculture [64]. (2) The coefficient of the direct effect of the level of investment in forestry fixed assets is significantly positive, and the indirect effect is negative but not significant, indicating that increasing regional investment in forestry fixed assets is beneficial to local forestry economic growth, and the spatial spillover effect is not significant. (3) The indirect effect of forest pest and disease control rate is significantly negative, while the direct effect is not significant, indicating that due to the mobility of pests and diseases, when local disaster control is carried out, the tree diseases will spread to the surrounding areas and thus reduce the economic growth of the surrounding areas. (4) The direct and indirect effects of the number of students in higher education are significantly positive,

indicating that human capital in higher education can not only promote local forestry economic growth, but also improve the forestry economic development in the surrounding areas. This indicates that human capital is the fundamental driving force to promote technology innovation and sustained economic growth and is an important initiative to promote forestry economic growth. (5) The direct and indirect effects of industrial structure upgrading are significantly negative, as faster regional industrial structure upgrading indicates that its secondary and tertiary industries are the main force. The more rapidly the local economy develops, the less capital will flow into the agriculture and forestry industries, thus making it difficult to promote the local forestry economic growth [61], showing a negative effect on the economic growth of the forestry industry in the region. (6) The direct effect of forest area is significantly positive, and the indirect effect is positive but not significant. The area of forest will affect the scale of the forestry industry, which in turn affects the output value of forestry. A larger forest area in a region could effectively promote the economic growth of the local forestry industry.

### 4.2.4. Industry Heterogeneity Analysis

Because of the different attributes of the three forestry industries, this paper further examines each of the three forestry industries separately. The spatial effects of forestry primary, secondary, and tertiary industries were analyzed under the geographic distance spatial weight matrix to clarify the impact of FTI on the economic growth of the forestry industry, and the regression results are detailed in Table 7.

**Table 7.** Regression results of spatial Durbin model for primary, secondary and tertiary forestry industries.

| Variable | Main | | | Total Effect | | |
| --- | --- | --- | --- | --- | --- | --- |
| | Primary | Secondary | Tertiary | Primary | Secondary | Tertiary |
| FTI | 0.0841 *** | 0.494 *** | 0.0977 *** | 0.273 *** | 2.630 *** | 1.061 *** |
| | (0.0123) | (0.0260) | (0.0110) | (0.0689) | (0.421) | (0.246) |
| Control variables | control | control | control | control | control | control |
| **Variable** | **Direct Effect** | | | **Indirect Effect** | | |
| | Primary | Secondary | Tertiary | Primary | Secondary | Tertiary |
| FTI | 0.0855 *** | 0.529 *** | 0.125 *** | 0.187 *** | 2.101 *** | 0.936 *** |
| | (0.0125) | (0.0283) | (0.0141) | (0.0660) | (0.408) | (0.237) |
| Control variables | control | control | control | control | control | control |

Note: *** represents significance level of 10%, respectively.

The regression coefficients of in FTI on the primary, secondary, and tertiary forestry industries are all significantly positive at the 1% level, which again verifies hypothesis 1: FTI promotes the development of forestry industry. From the spatial spillover effect, the direct and indirect effects of FTI on the three forestry industries are significantly positive, indicating that FTI can not only promote the local economic growth of the three forestry industries, but also promotes the economic growth level of the three forestry industries in the surrounding areas through the spillover effect.

### 4.3. Robustness Check
#### 4.3.1. Adjusted Sample Period

In order to further liberate and develop forestry productivity, develop modern forestry, increase farmers' income, and build ecological civilization, in 2008, China began to comprehensively promote the reform of the collective forestry tenure reform, implementing collective forest land management rights and forest ownership rights to farmers, establishing the status of farmers as the main business entity, further liberating and developing rural productivity, fostering market players in forestry development, and playing the fun-

damental role of the market in the allocation of forestry production factors [65]. This policy is conducive to increasing the quantity of forests, improving their quality, meeting the diversified needs of society for forestry, and promoting the development of modern forestry. Since the policy was implemented in 2008, it has improved the allocation efficiency of forest resources, activated the comprehensive productivity of forestry, and China's forestry industry has started to flourish. Therefore, the article uses a reduced sample period to explore the impact of FTI on the economic growth of forestry industry. The sample period of the model was chosen as 2008–2020, and other explained variables, explanatory variables, and control variables remain unchanged. The regression results are basically consistent with the previous paper, indicating that the estimation results are robust, as shown in Table 8.

**Table 8.** Results of robustness checks from 2008–2020.

| Variable | Main | Variables | WX |
|---|---|---|---|
| FTI | 0.216 *** | W FTI | 1.560 *** |
| | (0.0363) | | (0.172) |
| FEG | −0.0186 *** | W FEG | 0.0035 |
| | (0.0020) | | (0.0112) |
| ln FFL | 0.0051 ** | W ln FFL | −0.0049 |
| | (0.0021) | | (0.0139) |
| DPR | 0.0002 | W DPR | −0.0034 ** |
| | (0.0002) | | (0.0014) |
| ln EL | −0.0166 * | W ln EL | −0.0374 |
| | (0.0086) | | (0.0366) |
| UIS | −0.0040 *** | W UIS | 0.0009 |
| | (0.0011) | | (0.0042) |
| ln LA | 0.0446 *** | W ln LA | 0.0491 * |
| | (0.0043) | | (0.0282) |
| Spatial rho | 0.395 *** | Variance sigma2_e | 0.0034 *** |
| | (0.153) | | (0.0002) |

Note: ***, **, and * represent significance levels of 1%, 5%, and 10%, respectively.

### 4.3.2. Replacing the Spatial Weight Matrix

In order to ensure the robustness of the results, this paper replaces the geographic distance spatial weight matrix with the neighboring spatial weight matrix and the economic distance weight matrix for robustness testing and conducts the spatial Durbin model regression again. The regression results are basically consistent with the previous paper, indicating that the estimation results are robust, as shown in Table 9.

### *4.4. Threshold Effect Test*

To identify the specific form of the panel regression model, this paper uses the level of FTI as the threshold variable and uses the Bootstrap method to repeat the sampling 300 times to test the single threshold effect and the triple threshold effect, respectively, and obtain the statistical values F, *p*-value, and the corresponding critical values, as detailed in Figure 2 and Table 10. From the results, it can be seen that the single threshold effect test of FTI level is significant at the 1% level, and the two-three threshold effects are not significant, indicating that there is a non-linear characteristic of a single threshold effect of FTI level on the economic growth of the forestry industry, which initially verifies hypothesis 3.

**Table 9.** Robustness test results for replacing spatial weights.

| Variable | Main | | WX | | Direct Effect | | Indirect Effect | |
|---|---|---|---|---|---|---|---|---|
| | ED | AD | ED | AD | ED | AD | ED | AD |
| FTI | 0.137 *** | 0.0804 *** | 0.285 *** | 0.131 *** | 0.154 *** | 0.111 *** | 0.433 *** | 0.348 *** |
| | (0.0275) | (0.0252) | (0.0747) | (0.0410) | (0.0279) | (0.0275) | (0.100) | (0.0747) |
| FEG | −0.0083 *** | −0.0062 *** | 0.0289 *** | 0.0097 *** | −0.0071 *** | −0.0052 *** | 0.0362 *** | 0.0130 *** |
| | (0.0014) | (0.0012) | (0.0039) | (0.0021) | (0.0014) | (0.0012) | (0.0056) | (0.0041) |
| ln FFL | −0.0021 | 0.0052 *** | 0.0034 | −0.0026 | −0.0018 | 0.0054 *** | 0.0041 | 0.0007 |
| | (0.0018) | (0.0015) | (0.0049) | (0.0038) | (0.0018) | (0.0018) | (0.0066) | (0.0080) |
| DPR | 0.00003 | 0.0003 ** | −0.0006 | −0.0004 | 0.0000 | 0.0002 * | −0.0008 | −0.0005 |
| | (0.0001) | (0.0001) | (0.0004) | (0.0003) | (0.0001) | (0.0001) | (0.0006) | (0.0005) |
| ln EL | 0.0138 *** | 0.0140 *** | 0.0044 | 0.0104 | 0.0140 *** | 0.0170 *** | 0.0132 | 0.0377 ** |
| | (0.0052) | (0.0047) | (0.0136) | (0.0080) | (0.0052) | (0.0047) | (0.0195) | (0.0152) |
| UIS | −0.0032 *** | −0.0004 | 0.0035 * | 0.0016 | −0.0030 *** | −0.0001 | 0.0034 | 0.0028 |
| | (0.0006) | (0.0006) | (0.0019) | (0.0012) | (0.0006) | (0.0007) | (0.0024) | (0.0027) |
| ln LA | 0.0195 *** | 0.0267 *** | 0.0127 | 0.0049 | 0.0203 *** | 0.0301 *** | 0.0241 ** | 0.0390 *** |
| | (0.0026) | (0.0027) | (0.0083) | (0.0062) | (0.0026) | (0.0029) | (0.0114) | (0.0140) |

Note: ***, **, and * represent significance levels of 1%, 5%, and 10%, respectively; ED stands for economic distance, AD stands for adjacent distance.

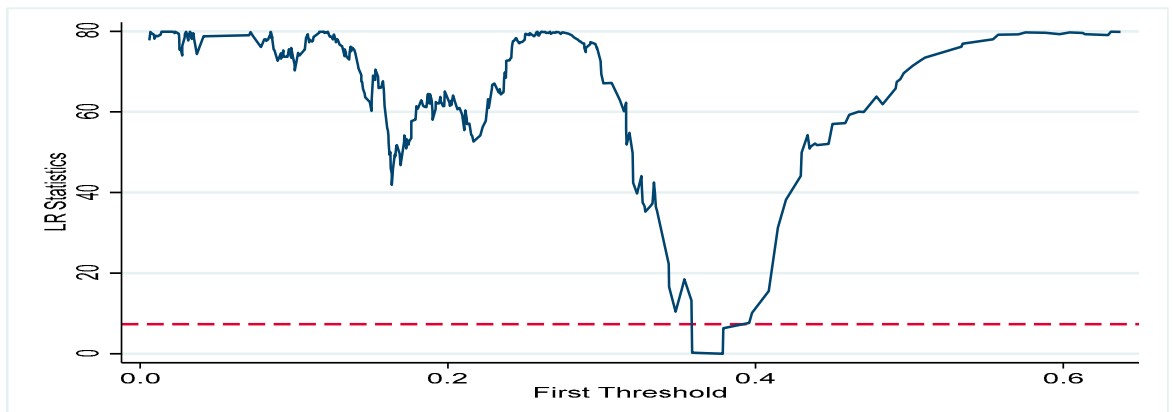

**Figure 2.** Threshold test based on FTI.

**Table 10.** Threshold effect test results.

| Test Parameter | Threshold | F Value | *p* Value | The Critical Value | | |
|---|---|---|---|---|---|---|
| | | | | 10% | 5% | 1% |
| FTI | Threshold 1 | 78.43 | 0.003 | 30.895 | 36.084 | 46.957 |
| | Threshold 2 | 14.83 | 0.470 | 22.608 | 25.519 | 33.201 |
| | Threshold 3 | 6.03 | 0.940 | 21.253 | 23.684 | 31.873 |

In order to better study the impact of FTI on the economic growth of the forestry industry, a single threshold model is selected for estimation in this paper, as detailed in Table 11.

When the level of FTI is less than 0.379, the coefficient of its effect on the economic growth of forestry industry is 0.642, and it is significantly positive at the 1% level. When the level of FTI is higher than 0.379, its impact coefficient on the economic growth of the forestry industry is 0.369, and it is significantly positive at the 1% level, which also indicates that when the level of FTI reaches a certain level, the promotion of economic growth of the forestry industry shows a trend of diminishing marginal benefit, which again verifies hypothesis 3. This also shows the level of FTI within a reasonable range in order to fully promote the impact.

**Table 11.** Threshold regression results.

| Variable | Coefficient | Variable | Coefficient |
|---|---|---|---|
| threshold | 0.379 | | |
| FTI(FTI ≤ 0.379) | 0.642 *** | FTI(FTI > 0.379) | 0.369 *** |
| | (0.0637) | | (0.0425) |
| FEG | −0.0063 *** | ln EL | 0.122 *** |
| | (0.0013) | | (0.0079) |
| ln FFL | −0.0018 | UIS | 0.0008 |
| | (0.0017) | | (0.0007) |
| DPR | 0.0007 *** | ln LA | −0.0026 |
| | (0.0001) | | (0.0136) |
| _cons | −1.680 *** | | |
| | (0.103) | | |

Note: *** represents significance level of 10%, respectively.

## 5. Discussions

This paper mainly uses the spatial Durbin model and threshold effect to analyze the impact and spatial effect of FTI on the economic development of forestry industry based on the data of 31 provinces in China from 2002 to 2020. The conclusions indicate that there is a spatial correlation between FTI and the level of economic development of the forestry industry in 31 provinces in China during 2002–2020. The level of FTI is higher in regions with faster economic development and rich forestry resources, but China has less forest area and a relatively weaker level of FTI compared to other countries, e.g., in Europe [9,66]. So, the overall level of FTI in China is uneven and needs to be improved. Local governments should increase the investment in forestry industry, improve the level of forestry technology innovation, and promote the development of forestry industry [67].

FTI has a significant role in promoting the forestry industry economy, and there is a spatial spillover effect that can promote the economic growth of the forestry industry in the surrounding areas, which is different from the findings of Chen N [68] et al. This may be due to the fact that with the improvement of innovation consciousness, each region pays more attention to the matching degree of the introduction of talents in technology talents to the regional forestry development, which makes FTI work to the maximum extent, improves the efficiency of forest resources utilization, and promotes the development of forestry industry. At the same time, there is a spatial autocorrelation of FTI level, which is consistent with Wei Huang's [69] research results. If the spatial factor is ignored, it will lead to biased results, so this study fully considered the spatial effect and enriched the research results of forestry economy based on Tan J et al. [70] to make our results more scientific. Besides, there is heterogeneity in the promotion effect of forestry industry, and the results show that forestry technology innovation has the most obvious promotion effect on the secondary industry. Although the level of forestry technology innovation has a spatial spillover effect on the economic growth of the forestry industry, but the level of economic development, resource endowment, and the number of scientific and technological talent in each region is different, and if the advanced technology and experience of other regions are blindly copied, it is likely to inhibit the development of the forestry industry economy in the region. Therefore, each region should use the spillover effect of FTI level according to the actual situation to promote the development of forestry industry [71].

In addition, FTI has a positive effect on the economic development of the forestry industry, but the relationship between the two shows a non-linear relationship. So, the level of FTI should be continuously improved within a reasonable range to promote the growth of the forestry industry. If the level of FTI exceeds this range, its role in promoting the economic growth of the forestry industry is weakened, and at this time there will be a social cost, such as waste of resources [72–75].

This paper does have certain flaws. On the one hand, the measurement method of FTI level is relatively simple, which is mainly due to the lack of data and methods of FTI level. On the other hand, the research on FTI is still incomplete and the related research

literature is less. Therefore, the choice of influencing variables can only be exploratory in combination with the relevant previous studies.

In future research, we will try to find micro data that better represent FTI indicators and further refine and classify FTI level indicators. We will also improve the methodology for measuring the level of FTI in order to obtain more accurate values. At the same time, we will also explore the impact of FTI on the forestry industry economy by controlling more variables that are necessary, thus better analyzing the impact of FTI level on the forestry industry development and improving effective suggestions for relevant policy makers.

## 6. Conclusions and Suggestions

### 6.1. Conclusions

This paper uses the spatial Durbin model to empirically test the spatial spillover and threshold effects of FTI on the economic growth of the forestry industry based on panel data of 31 provinces in China from 2002 to 2020, and further explores the spatial impact analysis on the three major forestry industries. The main findings are as follows.

(1) From the perspective of factors affecting the level of FTI on the economic growth of the forestry industry, the training of talent in the field of forestry should be increased to improve the soft power of forestry industry development, enhance the implementation of FTI research and results through the introduction of outstanding talents, and then improve the level of FTI. The investment in regional forestry fixed assets should be strengthened to improve the hard foundation of forestry industry development. Local government departments should reasonably plan the ratio of government financial expenditures, strengthen the efficiency of capital utilization around the world, and avoid affecting the development of the forestry industry economy due to excessive intervention and reliance on financial funds.

(2) From the spatial spillover effect of the level of FTI on the forestry industry, the exchange and learning system between each region should be strengthened to promote technical cooperation in forestry and high-quality development of the regional forestry industry.

(3) From the non-linear relationship between FTI on forestry industry growth, China should avoid diminishing marginal benefits when improving the potential of FTI for the economic growth of the forestry industry, play to the strong advantage of FTI, and actively promote China's forestry industry to the middle and high end of the global value chain.

Through this study, we hope to provide policy-making information for local governments in China to formulate forestry industry policies, and at the same time, to provide references for the high-quality development of forestry industries in regions with similar situations in the world.

### 6.2. Suggestions

Based on the above research, the following suggestions are put forward to promote the healthy development of the forestry industry economy.

(1)　FTI is an important driving force for the economic development of the forestry industry, and the level of FTI should be gradually improved. From the input point of view, we should strengthen the investment in regional forestry and introduce more talent to the forestry industry to enhance the results of relevant scientific research.

(2)　Communication and learning systems should be strengthened among regions to achieve technical cooperation and association in order to exploit the strong advantages of FTI. At the same time, the flow of forestry talent and funds to areas with slow economic development but rich forestry resources should be encouraged to reduce the technical barriers between regions and promote the synergistic development of FTI level between regions, thus forming a unique FTI development model, and promoting the high-quality development of regional forestry industries.

(3)　Since the effect of forestry technology level on the economic growth of the forestry industry shows a non-linear correlation, economic benefits should also be taken into

account to measure the regional input–income ratio while enhancing the regional FTI level.

(4) Financial investment in the forestry industry should be increased at all levels, fully utilizing the guiding role of financial funds, enhancing the potential of forestry technology innovation for the economic growth of the forestry industry, and actively promoting the high-quality development of China's forestry industry.

**Author Contributions:** Conceptualization, W.C. and Z.M.; methodology, Z.M. and Z.L.; software, Z.M.; vali-dation, Z.M., W.C. and T.L.; formal analysis, Z.M.; investigation, Z.M. and W.C.; resources, W.C.; data curation, Z.M., J.L and W.C.; writing—original draft preparation, W.C. and Z.M.; writing—review and editing, J.L. and Z.L.; visualization, J.L. and Z.L.; supervision, W.C.; project administration, W.C.; funding acquisition, W.C. All authors have read and agreed to the published version of the manuscript.

**Funding:** This research was funded by the National Forestry and Grassland Administration project "2021 Forest Products Market Research" (Grant No. JVC-2022-0010).

**Institutional Review Board Statement:** Not applicable.

**Informed Consent Statement:** Not applicable.

**Data Availability Statement:** Not applicable.

**Conflicts of Interest:** The authors declare no conflict of interest.

## Appendix A

**Table A1.** Forestry industry and specific classification.

| Forestry Industry | Specific Classification |
| --- | --- |
| Forestry primary industry | Cultivation and planting of forest trees<br>Timber and bamboo harvesting and transportation<br>Cultivation and collection of economic forest products<br>Flower planting<br>Breeding and utilization of terrestrial wildlife<br>Forestry production support services |
| Forestry secondary production classification | Wood processing and wood, bamboo, rattan, palm, reed products manufacturing<br>Wood, bamboo, rattan furniture manufacturing<br>Wood, bamboo, reed pulp and paper<br>Forestry chemical products manufacturing<br>Wood, bamboo crafts and educational and sporting goods manufacturing<br>Non-wood forest products processing manufacturing<br>Other |
| Forestry tertiary industry classification | Forestry tourism and leisure services<br>Forestry ecological services<br>Forestry professional and technical services<br>Forestry public management and other organizational services<br>Forest products, fruit and tea, wood and bamboo and their processing products wholesale and retail trade<br>Other |

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
