# Peer review of "Spatial Effect Analysis of Forestry Technology Innovation on Forestry Industry Economic Growth"

_forests, doi:10.3390/f14030557_

Round 1
Reviewer 1 Report
The effect of forestry technology innovation on the economic growth of the forestry industry is analyzed using a spatial dependence model. I think it is a valuable paper in terms of research necessity and differentiation. However, to improve the completeness of the paper, please read the following comments and revise the paper.
<Title>
Since this paper analyzes the causal relationship that forestry technology innovation has on the economic growth of the forestry industry, the title should be changed to reveal this causal relationship analysis. For example, "Spatial Effect Analysis of Forestry Technology Innovation on Forestry Industry Economic Growth"
<Introduction>
The research purpose is not clearly presented in the introduction. Currently, there are only research questions and academic contributions. Please indicate the research purpose and key research contents.
(Line 111) I wish I could cut the paragraph off here.
<2. Theoretical Mechanism>
(Line 132) Why is this chapter called "theoretical Mechanisms"? "Research questions, or "Research hypothesis" would be more appropriate.
(Line 134-135) Incomplete sentence. Need clarification.
(Line 139) "the achievements of forestry technology ..." -> "the achievements of forestry technology innovation..." ???
(Line 176) It should be clarified in which aspect the "threshold effect" is meant.
(Line 191) The basis for the hypotheses for regional differences has not been clearly explained above.
<3. Data Sources and Methodology>
(Line 218) The order of the descriptions in the Table 1 must match the order of the preceding descriptions.
(Line 219) Data structure (panel data?) and temporal and spatial scope are not clearly presented.
(Line 234) Moran's I index is not sufficient to test for spatial dependencies. Spatial LM test for panel model needs to be considered.
(Line 252) Explanation of panel model specification is also needed.
(Line 264, 281) Position the equation numbers appropriately.
(Line 289) Lack of information on missing data. Interpolation or extrapolation of missing data should be done with great care. It is necessary to clarify which method was used and review how sensitive the method is.
(Line 291 and below) Use of ":" or incomplete sentences. Does the description of each variable fit the format of this journal?
(Line 321) Need to insert units.
<4. Results>
(Line 357, 389) Titles in Tables 4 and 5 need to be changed. Same as Table 2.
(Line 414) This seems like an unnecessary analysis... Regional differences can also be considered in the panel model. If a model that is separated by region is to be estimated, the hypothesis that each region is spatially independent of each other and the hypothesis that each region has different coefficients must be tested.
(Line 478) Robustness checks using other spatial weight matrices are inappropriate. The selection of an appropriate spatial weight matrix among several available matrices should have already been considered in the model selection process in the first place.
<5. Conclusions and suggestions>
Please make sure that the numbering of each paragraph in the conclusion conforms to the format required by this journal.
Reviewer 2 Report
Duplicate sentence on line 96.
I worry that economic activity related to forestry is used both to construct the index of investment that serves as the primary independent variable, and as the dependent variable in the regressions. Is there a time lag or other difference so that the same numbers are not on each side of the equations?
Author Response
请参阅附件。

Reviewer 3 Report
# Review of first version
Good that you clarified your research question lines 117 to 120. The 3 hypotheses
expressed pages 4 and 5 are clear, but your introduction is messy and confusing, in
particular it lacks a definition of forest industries. The spatial spill over effect,
crowding out effect and threshold issues would benefit from being introduced in a
clearer way at the beginning of your article.
Main comments about the content:
- I think the forest area of each region should be added as an additional control
variable in the models.
- The authors should define the boundaries of the forest sector under consideration.
Please describe precisely what type of economic activities are included in forestry
technology innovation?
- I suggest explaining in the introduction that the core of your analysis is based on a
measure of technological innovation, based on previously published works. Explain what
is the contribution of those work. Looking at citations [32, 36-38] 2 seem to be about
forestry and 2 about other sectors.
- Your dataset has a time component. Could you provide insights on the speed of
innovation? Which regions have increased their innovation faster than others?
Language and proof reading:
- The article would have benefited from more proof reading. There are repeated sentences
that should have been removed. A table title was repeated 3 times for tables 2, 4 and 5.
The same variable has different names through the article (explain why or harmonize
the variable names).
- The expression "forestry technology innovation" is repeated 94 times in the document.
Please have pity of the reader and use the acronym FTI or remove excessive
duplications.
Lines 35-36,
> "23.90 billion yuan in 1949 to 81176 billion yuan in 35 tional affiliations. 2020, the
> total output value of forestry increased 3,396 times;"
The 1949 value should be adjusted for inflation in order to be comparable to the 2020
value. A comparison in terms of material footprint, would be of great interest to the
reader, could you also provide values in terms of cubic meters of wood produced?
Line 50
> "forestry construction"
What do you mean by forestry construction? Please clarify. Does this concern the
construction of new industrial capacity? Or wood-based construction? Or something else?
Line 52
> "For a long time, China's forestry development mainly relies
English language suggestion:
relied
Line 66
> after eliminating input factors,
Can you clarify what are those input factors?
Lines 60 to 80 are redundant and could be shortened. Part of these text is already
expressed before and repeated afterwards. For example the following sentence could be
removed completely:
> "With the increasing resource and environmental constraints in recent years, the
> contribution of technology innovation to forestry economic growth has become
> increasingly obvious, especially in the current critical period of accelerating the
> high-quality development of forestry, enhancing the level of forestry technology
> innovation will undoubtedly play an equally significant role in improving China's
> forestry technology innovation ability and promoting rapid forestry economic growth"
Lines 79-81
> Forestry technology innovation can exploit new technologies and methods to achieve
> improvements in seeds, fertilizers, and mechanical equipment, which in turn can
> improve productivity development
It is important to put innovation in context and to explain what technologies will be
impacted. The article should define the boundaries of the sector under consideration.
What do you consider as forest technology? Does it include tree planting and thinning
operations? Does it include forest products harvesting operations? Does it include
sawmills? Does it include the paper industry? Does it include the furniture industry?
Line 142 mentions "farmers", and the paragraph above talks about "seeds" so I suppose
your analysis focuses only tree planting, thinning and maybe harvesting operations.
However lines 463-467 and table 8 distinguish primary, secondary and tertiary forest
industries, please define these.
Please also clarify whether you focus only on plantation forests or also on naturally
re-generated forests. Forest characteristics can have a great influence on the type of
products that can be made and therefore on the level of innovations required to improve
productivity. It would help the reader to understand forest characteristics in
particular: what is the forest area? What are the main species? What is the average
forest age in each of the regions of interest? Could you provide such descriptive
statistics?
LIne 84
> "reduce the risk of economic growth caused by natural disasters."
Language:
reduce the risk **to** economic growth
Lines 96 to 99
> "Reviewing the literature, scholars at home and abroad pay more attention to the
> optimization and adjustment of forestry industry structure, forestry industry
> performance evaluation, forestry production efficiency, forestry industry
> agglomeration and high-quality development of forestry industry [14-17],"
Remove this entire sentence, it is a repetition of the previous one.
Lines 105, 107
>"explored the impact of technological innovation on human well-being and global
>sustainable development from an ecological perspective based on the perspective of
>technological innovation."
Please remove the last repetition of "technological innovation" in the same sentence.
Lines 117 to 120
> "At the same time, what has the impact of forestry technology innovation on forestry
> economic growth when spatial spillover is considered? How to effectively use the
> positive impact of forestry science and technology innovation to achieve sustainable
> growth of forestry economy?"
Good that you have those research question clearly labelled here.
Language suggestion:
what is the impact
Lines 124-126
> "In terms of research content, technology innovation is precisely applied to the
> forestry field, to explore its impact on the forestry industry economy and the
> marginal contribution changes..."
What method do you apply here? Technology innovation is a phenomenon, it's not a
research method. Should should specify here what econometric model you have used.
Line 131:
> "makes up for the shortcomings of existing studies."
How? You already mentioned the lack of treatment of spatial spill over effects. It would
be nice to already give a hint at what are the shortcomings.
Line 134
> "Technology innovation is the first productivity.
This sentence is too short. Maybe you want to say it's a factor influencing
productivity?
Line 139 - 141
> t. First, the achievements of forestry technology can improve the types of forest
> products, enhance the quality of forest products, and im- prove labor efficiency
Lines 205-216
> "... combining the research contents of previous scholars [32, 36-38], this paper
> constructs an evaluation index system of forestry technology innovation based on three
> dimensions of innovation base, innovation input and innovation output, which mainly
> includes three aspects.(1)The foundation of forestry technology innovation, that is,
> the economic and social development status, human capital and other material
> infrastructure and related facilities of the region where the main body of forestry
> technology innovation is located, providing key support for technological
> innovation.(2)Technology innovation input: that is, the full application of various
> innovative knowledge and resources in forestry technology activities and technology
> transformation, is an important part of the integration and distribution of the use
> of technology innovation. (3) Technology innovation output: that is, the part of the
> output and benefit increase of forestry technology innovation achievements, which
> can test the results and productivity levels of forestry technology innovation, and
> is the final result of forestry technology innovation (Table 1)."
Line 233
> Figure 1. Forestry technology innovation levels of 31 provinces in 2002, 2011 and 2020.
To facilitate interpretation, order regions from the highest to the lowest level of
innovation in 2020, or by the mean value as you prefer.
How do you explain the fact that large industrial regions of Shanghai and Beijing have
decreasing levels of forestry technological innovation through time i.e. levels in 2020
are lower than levels in 2002? Apart from the vast forest resources, what is particular
with Guangdong in terms of forestry innovation?
It would be interesting to provide statistics in terms of forest area by region as a
supplementary material. If possible separating plantation forests from naturally
regenerated forests. In fact the forest area should be used as an additional regressor
in your model, if there are region with large protected natural forests, it might also
be important to get the forest area available for wood supply as you are focusing on the
industrial forest products.
Line 263
> "SAR model."
The definition of SAR should be given here. It's given later at line 354.
The definition of SEM and SDM should be given here as well.
Line 289-290
> "The missing data are estimated by simple linear extrapolation method and
> interpolation method."
What is the amount of missing data per region, can you add this to the descriptive
statistics? Is the linear interpolation likely to impact your results?
Line 291
> "Explained variable. Economic growth of forestry industry (EGFI):"
Remove or make it a sentence.
Line 196
> "The level of forestry technology innovation (FTI). In the previous literature,
> scholars have used the number of patent applications, authorizations and sales revenue
> after new product development to measure."
Suggestion:
In previous literature, scholars have used the number of patent applications,
authorizations and sales revenue after new product development to measure the level of
forestry technology innovation (FTI).
Line 302
> "Financial expenditures of the government(govex):"
You later call this variable "fiexpend" in result table 5 for example. Please harmonize
to use one name otherwise it's confusing for the reader.
Lines 306, 307
> "Forestry fixed asset level(ffixinve): The level of investment in forestry fixed
> assets"
What are included in fixed assets? Land, machinery, industrial facilities? This is in
particular related to my earlier questions about defining the boundaries of the system.
What is the level of vertical integration of the forestry industry in China. Are the
farmers independent of the downstream processing industries, sawmills and paper mills?
Line 321
> "Table 2. Descriptive statistics of variables."
govex appears twice in the table.
- Forest area should be added as an additional regressor in your model.
- Displaying the most recent EFGI and FTI values on a choropleth map of forest area by
region would help understand the spatial repartition of those variables and the
correlation with forest area.
Line 352
> "the great likelihood method"
I think it is called the maximum likelihood method.
Line 357
> Table 4. Descriptive statistics of variables.
This title was already used for table 2. Table 4 should have another title.
> Table 5. "Descriptive statistics of variables."
This title was already used for table 2. Table 5 should have another title.
How is it possible that the R2 have the same value of 0.187 for the two models? Was
there a mistake when copying the results?
One variable starts with "ln": lnstudent. Does this mean that you took the natural
logarithm of that variable? What about the other variable? Would it make sense to
compute the model with all variables expressed in logarithm? This is frequently done in
macroneconomic models.
> Table 6 Marginal effect decomposition of spatial Durbin model.
All models have a very low R2 equal to 0.033. What are the consequences of such a low
R2?
Lines 415-418
> "The differences in resource endowment, development level and social environment
> policies of different regions will make the level of forestry technology innovation
> and for- est industry economic growth show heterogeneity, so it is necessary to
> conduct spatial heterogeneity analysis."
You should also capture the resource endowment by adding forest area as a control
variable.
Lines 453-455
> "However, the progress of forestry technology innovation in the region will generate
> tech- nology spillover and thus drive the economic development of forestry industry in
> neigh- boring regions."
What about water precipitations and soil quality? Are there enough areas suitable for
forest plantations and/or naturally regenerated forests in the western regions?
Lines 463-467
> "Because of the different attributes of the three forestry industries, this paper
> further examines each of the three forestry industries separately. The spatial effects
> of forestry primary, secondary and tertiary industries were analyzed under the
> geographic distance spatial weight matrix to clarify the impact of forestry technology
> innovation on the eco- nomic growth of forestry industry, and the regression results
> are detailed in Table 8."
Please explain what are these 3 sectors.
LInes 483 Table 9 "Table 9. Results of robustness check."
Please explain the results of the WX and indirect models which seem to indicate that FTI
is not significant anymore.
Lines 519-521
> "the overall level of forestry technology innovation of 31 provinces in China during
> 2002-2020 is uneven and needs to be improved."
It is strange for a scientific article to be so prescriptive. Replace by "can be
improved".
Lines 533-535
> "When increasing investment in fixed assets can expand production capacity,"
I would really like to know what are these fixed assets? What industries are we talking
about here? The expression "industrial structure" seem to refer to large industrial
facilities, do you mean sawnmills and pulp milss or is this still only about forestry
machinery?
> "Financial support of agriculture inhibits the economic development of forestry
> industry in the local and surrounding areas, indicating that the government's economic
> interventions hinder the economic growth of forestry industry in the local and
> surround- ing areas."
- You mention "social and environmental policies" (lines 564-565). What is the nature
of the government financial support? Does it have a link with protected areas, such as
national parks? In which case you should distinguish productive forests from forests
which have other purposes such as biodiversity protection or tourism. Some o these
areas would be run at a loss by governments because forests provide positive
externalities to society which are not captured in market values.
- If this is the case, it would be important to split the financial support related to
nature restauration (point above) from the financial support related to subsidies to
farmers. After the split, if the negative correlation still holds, this would then
make your conclusion stronger.
Lines 588-590 clarify that government support seems to be more about subsidies to the
industry or direct ownership by the state of forest entreprises and less about support
to other forest functions (recreation, water protection, biodiversity habitat).
> "China's forestry sector is mostly state-owned enterprises, which rely on financial
> support to a certain extent. The efficiency of funds utilization in different places
> should be strengthened to prevent local over-reliance on financial funds,"
Reviewer 4 Report
Dear authors,
First of all, many congratulations on the topic addressed in this research study, it is, in fact, very interesting. However, there are two aspects that can still be improved so that the study can be properly published, namely:
-The number of bibliographic references can and should be increased;
- In the conclusion, the authors went into detail, which is very good, but I do not understand the enumeration of conclusions.
Reviewer 5 Report
Dear author(s),
there are some inspiring insights thorough the manuscript and I tend to agree on its publication. However, there are few points that needs to be quickly addressed to improve its overall communication:
Title:
1/ indicate what was revealed by you analysis
Abstract:
2/ better address our international audience, do not indicate local impact
3/ strictly follow the established schema of writing academic Abstract: A/ introduction (urgency and significance of the research hypothesis); B/ principles of the methods used + key results; C/ conclusions (commercial and environmental impacts)
4/ novelty and significance of this work is poorly communicated, clearly indicate how will humanity benefit from these revelations (quantify the environmental and industrial importance)
Introduction:
5/ you can use China as a case study, however, the introduction to the topic should be carried out from a greater perspective (global point of view)
6/ use any freely convertible currency to quantify financial values so that our international audience of readers can better understand your text (refer to papers "The Influence of the International Price of Oil on the Value of the EUR/USD Exchange Rate" and "Predicting future Brent oil price on global markets")
7/ please understand that our readers outside China cannot understand terms like "opening up of China"; "Five-Year Plan" etc., make sure that this chapter fully introduces any reader into to the topic, explain all the terms, units, abbreviations, Latin and Greek letters, and the whole context that is necessary for anyone (including experts from other disciplines) to understand the following chapters
8/ deeper review the latest trends in wood utilization and commercialization, refer to papers "Techno-economic analysis reveals the untapped potential of wood biochar" and "Clusters in Transition to Circular Economy: Evaluation of Relation"
9/ go straight to the point and more in depth, write more technically (always provide corresponding numbers), significantly condensate all the text by reducing ballast phrases and cliché
10/ modern economic theories of production and processing should not be ignored (refer to papers "Sustainable Organizational Performance, Cyber-Physical Production Networks, and Deep Learning-assisted Smart Process Planning in Industry 4.0-based Manufacturing Systems" and "Artificial Intelligence Data-driven Internet of Things Systems, Real-Time Advanced Analytics, and Cyber-Physical Production Networks in Sustainable Smart Manufacturing")
11/ the research hypothesis could be stated more clearly, condensate the research hypothesis into 1 short statement or question that will be subsequently confirmed or refuted, make sure the urgency and significance of the research hypothesis was justified in its environmental - economic nexus
Theoretical Mechanisms:
13/ lay the foundations of the methods that can be used to confirm or refute the research hypothesis
Data Sources and Methodology:
14/ the method must be presented in such a way that it can be reproduced anytime, by anyone, anywhere (do not create obstacles like referring to specific location etc.)
15/ please understand that the methodology must be described in a completely unambiguous way that does not allow for multiple interpretations (everyone who reads this chapter should get very precise instructions on how to repeat your procedure to achieve exactly the same results)
16/ make sure all the data is freely and easily available to our international audience or readers (in English)
17/ Fig. 1: consider removing all the local names, this brings nothing to our global audience
Results:
18/ avoid data overkill, present only the most most industrially important results
19/ each Tab. and Fig. should be provided with caption that describes A/ what can be seen and B/ how is this relevant to the research hypothesis
20/ Fig. 2: provide corresponding units to the X and Y axes
Conclusions and Suggestions:
21/ show more self-criticism to your work (can all the methods and results be fully trusted? what are the weaknesses of the methods used? where do the main measurement inaccuracies arise? what are the limitations from a commercial point of view? are the lessons learned transferable to other fields?)
22/ propose some improvements and direction for future research, refer to papers "Sustainable Industry 4.0 Wireless Networks, Smart Factory Performance, and Cognitive Automation in Cyber-Physical System-based Manufacturing" and "Impact of FDI in Economic Value Added: Empirical Study in Terms of Renewable Natural Resources Mining within Wood-processing Industry"
23/ reveal the main driving mechanisms of your results, provide deeper synthesis and reveal some more original/significant findings
24/ compare your results in more depth with the existing literature, identify the main deviations and try to explain the mechanisms by which they may have been caused (refer to papers "Energy Cycles: Nature, Turning Points and Role in England Economic Growth from 1700 to 2018" and "A Comprehensive Bibliometric Analysis of the Energy Poverty Literature: From 1942 to 2020")
25/ please understand that the Conclusion chapter is not a summary of your work, present only original and industrially significant revelations that have the potential to expand the horizon of human knowledge (higher level of generalization is mandatory)
Round 2
Reviewer 1 Report
It is thought that the author has accommodated my revision request as much as possible, and I believe that the completion of the paper has been improved. Thank you for the author's efforts to revise the paper.
Reviewer 3 Report
Dear authors, the addition in version 2 improve the paper. I have a few additional points below.
> "The forestry industry is a material production and ecological construction project that protects,
cultivates, manages and utilizes forest resources, and provides forest products as well as forest
services to society [1]."
Nice high level description. However this doesn't say if you include only primary
products harvest, or also secondary production. Luckily, further down in your answers, I
see that you described the specific classification. You explain in answer 45 that:
> "[...] the forestry industry in this paper refers to the sum of the primary, secondary
> and tertiary forestry industries classified according to the classification standard
> of China Forestry and Grassland Statistical Yearbook.
And in your answers to response 3, 9, 13, 24, 41 and 45 you paste the classification.
> The specific classification is as follows:
> Forestry first production classification [...]
> Forestry secondary production classification [...]
> Forestry tertiary industry classification [...]
The full classification is crucial to understand the scope of the paper. I recommend
that you add this classification as a table inside the paper, or at least as an annex to
the paper. If the classification was present in the article directly I would be a happy
reader.
Language suggestion: "first production" could be replaced by "primary production" which
is the term generally used in this case.
Response 24
> the proportion of forest area in each region accounting for less than 1% of the
> national forest area is 19.4%.
1% of national forest area is ambiguous in your sentence. Does it mean that these
regions cumulatively represent 1% of the forest area? Then what does the 19.4% mean? I
don't understand. Please reformulate. If I may make a suggestion, you could for example
illustrate the distribution by talking about the number of regions needed to reach 90%
of China's forest area. You could also illustrate the repartition within each region by
splitting them in three groups such as: A regions have more than 50% of their land area
covered by forests, B regions have between 5 and 50% of their land area covered by
forests and C regions have less than 5% of their land areas covered by forests. Replace
A, B and C by numbers of regions.
> Response 37: Thank you for your kind comment. We take logarithms of some variables in
> the paper, in order to improve the smoothness of the data and make it compatible with
> the fluctuation level of other variables, and also to eliminate the problem of
> heteroskedasticity of the data, moreover, the nature and correlation of the data will
> not be changed after taking logarithms.
In table 5 and 6, two variables start with "ln" (ln FFL, ln EL) while other variables
don't (FTI, FEG, DPR, UIS, LA). I presume that variables starting with "ln" are the one
where you compute the logarithm, while the others are not. Please explain why you made
that choice.
I wish you a good continuation.
